# Few Shot Hematopoietic Cell Classification

**Vu Nguyen**[*][1]                                                   VHNGUYEN@CS.STONYBROOK.EDU

**Prantik Howlader**[*][1]                                      PHOWLADER@CS.STONYBROOK.EDU
**Le Hou**[1]                                                              LEHHOU@CS.STONYBROOK.EDU
**Dimitris Samaras**[1]                                           SAMARAS@CS.STONYBROOK.EDU

**Rajarsi Gupta**[2]                                  RAJARSI.GUPTA@STONYBROOKMEDICINE.EDU

**Joel Saltz**[2]                                         JOEL.SALTZ@STONYBROOKMEDICINE.EDU

[1] *Stony Brook University, Department of Computer Science, USA*

[2] *Stony Brook University, Department of Biomedical Informatics, USA*

**Editors:** Accepted for publication at MIDL 2023

## Abstract

We propose a *few shot learning* approach for the problem of *hematopoietic cell classification* in digital pathology. In hematopoiesis cell classification, the classes correspond to the different stages of the cellular maturation process. Two consecutive stage categories are considered to have a neighborhood relationship, which implies a visual similarity between the two categories. We propose RelationVAE which incorporates these relationships between hematopoietic cell classes to robustly generate more data for the classes with limited training data. Specifically, we first model these relationships using a graphical model, and propose RelationVAE, a deep generative model which implements the graphical model. RelationVAE is trained to optimize the lower bound of the pairwise data likelihood of the graphical model. In this way, it can identify class level features of a specific class from a small number of input images together with the knowledge transferred from visually similar classes, leading to more robust sample synthesis. The experiments on our collected hematopoietic dataset show the improved results of our proposed RelationVAE over a baseline VAE model and other few shot learning methods. Our code and data are available at https://github.com/cvlab-stonybrook/hematopoiesis-relationvae.

**Keywords:** Hematopoiesis cell classification, Few Shot Learning, Variational AutoEncoder

## 1. Introduction

We propose a novel few-shot method for classification of hematopoietic cells. In hematopoiesis, cell classes represent different stages of the cell evolution process. For some stages, it is difficult to collect a large training dataset to train a recognition model. The proposed few shot method incorporates relationships between hematopoiesis cell categories to robustly generate more training samples for training a hematopoietic cell classification model.

Hematopoiesis is the development of specialized blood cells from stem cells in the bone marrow. In this process, maturing normal blood cells trade self-renewal properties for specialized functions. Hematopathologists carefully examine lineage-specific morphologic features of blood cells that evolve through sequential stages of maturation to evaluate hematopoiesis. During laboratory workup, the goal is to identify morphologic changes

---

[*] Contributed equally

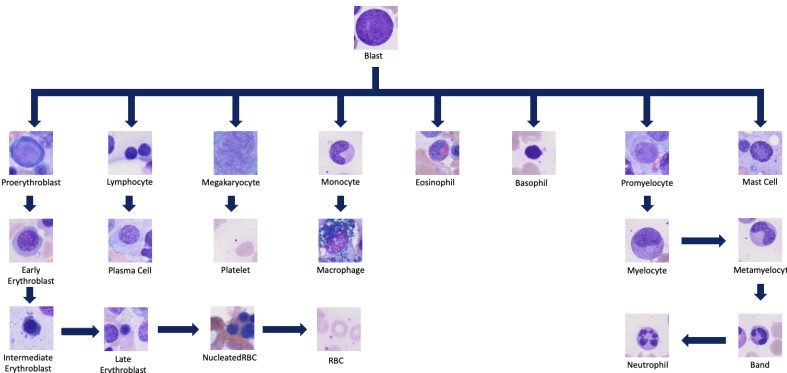

Figure 1: A conceptual example of the Hematoipoesis maturation process.

and/or abnormal cells associated with diseases such as anemia, myelodysplastic syndrome, and leukemia which is a very challenging task (Lee JY, 2020; Ye F, 2017; de Haan and Lazare, 2018), leading to the necessity of a machine learning method for hematopoietic cell classification. However, due to the rarity of some hematopoiesis categories, collecting a large training set is challenging and sometimes impossible.

We posit that the relationships between hematopoiesis categories could be used to mitigate the problem of limited training data. Specifically, each stage in the hematopoiesis process corresponds to a cell category. Two consecutive stages categories are considered to have a neighborhood relation, which implies a visual similarity between the two categories. Figure 1 shows an example of the evolution process starting with a multipotential hematopoietic stem cell. As a result, neighbor categories tend to be visually similar, for example Myelocyte and Metamyelocyte, Late Erythroblast and Nucleated RBC. It is therefore more robust to train a recognition model for rare classes if we refer to the appearance of their neighbors.

In this paper, we introduce a generative model that synthesizes training data for the hematopoiesis classes which have limited training data, utilizing prior class relationship knowledge. Specifically, our generative model is able to extract useful representations for novel classes (testing classes with limited training data) from a few input samples together with the knowledge transferred from their neighboring base classes (training classes with many training data), and generate training samples for the novel classes. Finally, generated samples are used to train classifiers for the novel classes.

To leverage this similarity information, we build a graphical model which represents the data generation process together with the visual similarity relationships. In this graphical model, we establish connections between neighboring classes (classes annotated to be visually similar) thus enabling knowledge transfer from a class to its neighbors. We derive the Pairwise Evidence Lower Bound (P-ELBO), the lower bound of the pairwise data likelihood on the graphical model. Based on this, we propose RelationVAE, a deep generative model which implements the graphical model. RelationVAE is trained to maximize the P-ELBO of the training data. As a result, the RelationVAE can generate data from a few training samples of a class utilizing the knowledge transferred from its neighbors. This data generation process is more robust compared to using only training samples without additional information from neighbor classes.

We evaluate the proposed method on a Hematopoiesis dataset that consists of 21 cell categories and relationships between them. Compared to baseline state-of-the-art methods, RelationVAE consistently achieves the best prediction performance on 1-shot setting and outperforms most baseline methods on 5-shot setting.

Our main contributions can be summarized as follows:

- We introduce a new Few-Shot learning paradigm which considers class relationships available in Hematopoiesis Cell Classification.
- We propose a graphical model for synthetic training data generation, which incorporates predefined similarity relationships between classes.
- We derive a Pairwise Evidence Lower Bound (P-ELBO) of the pairwise data likelihood on the graphical model and propose a deep generative model, RelationVAE, to optimize this lower bound. RelationVAE allows synthesizing samples from the novel class' few training samples, utilizing the knowledge transferred from its neighbor class.
- We evaluate the proposed method on a Hematopoiesis dataset that consists of 21 cell categories and relationships between them. RelationVAE consistently achieves better prediction performance compared to other state-of-the-art methods used as baselines.

## 2. Related Work

Few-shot learning can be formulated as a metric learning problem: learning a good feature embedding which maximizes the inter-class distances and minimizes the intra-class distances between samples (Xu et al., 2021; Vinyals et al., 2016; Tokmakov et al., 2019; Wertheimer et al., 2021; Rizve et al., 2021; Afrasiyabi et al., 2022; Xie et al., 2022). Another FSL approach is meta-Learning which aims to simulate the testing few-shot environment during the training phase to make the models familiar with this setting. One popular technique is to create different synthetic few shot classification tasks from the large training data to train the model, so that the model will be able to learn efficiently from a few training samples during testing time (Finn et al., 2017; Andrychowicz et al., 2016; Santoro et al., 2016; Ravi and Larochelle, 2017; Kim et al., 2019; Tang et al., 2021; Zhang et al., 2021). The third FSL approach is Data-Synthesis-based methods which directly address the problem of data scarcity by hallucinating training data of rare classes (Luo et al., 2019; Hariharan and Girshick, 2017; Wang et al., 2018; Zhang et al., 2018; Liu et al., 2019; Gao et al., 2018; Li et al., 2020; Ding et al., 2022; Hong et al., 2022b). LoFGAN (Gu et al., 2021) fuses multiple novel training images to generate additional images. DeltaGAN (Hong et al., 2022a), on the other hand, does not require multiple novel images during testing but proposes to learn intra-category transformation between images in the same category. During testing, these transformations are predicted from one or a few novel training images and are then used to generate more realistic images of the same class.

A few FSL methods incorporate **side information** to alleviate the problem of training data shortage. Yao et al. (Yao et al., 2020) enforce similarity relationship information between individual samples. Peng et al. (Peng et al., 2019) proposes a network that directly generates classifier weights for novel classes given a class similarity matrix together with class label embedding. In (Li et al., 2019) and (Liu et al., 2020), class hierarchies are employed to transferred knowledge from base classes to novel classes in the same group.

Shi et al. (Shi et al., 2020) utilize Graph Convolutional Neural Networks to incorporate the provided similarity affinity matrix. However, the type of relationship between hematopoiesis classes that we are considering in this paper are not integratable in these methods.

FSL is also attracting attention in the histopathology domain. (Walsh et al., 2022) conducts comprehensive experiments on applying and analyzing existing few shot Learning methods on human cell datasets. (Li et al., 2020) extends ProtoNet into 3D for Few Shot classification of macromolecular structures in Cryo-electron tomography. Many other histopathology FSL methods (Shakeri et al., 2022; Medela et al., 2019; Deuschel et al., 2021) focus on few shot domain transfer in which some of the domains (tissue type) have limited training data. Specifically, (Medela et al., 2019) and (Deuschel et al., 2021) learn a feature extractor from base classes on one tissue type, and train a simple classifier for novel classes on another tissue type. In summary, these histopathology FSL methods are based on simple metric learning techniques (similar to MatchingNet and RelationNet) which are shown to be inferior to our method in Sec. 4.

## 3. Proposed Method

### 3.1. Few Shot Learning with Visual Similarity Information

The conventional few-shot learning setting contains two sets of data: (i) base data $D_{base} = \{(x, y)\} \subset X \times Y_{base}$ and (ii) novel data $D_{novel} = \{(x, y)\} \subset X \times Y_{novel}$ in which $X$ is the set of samples and two set of classes $Y_{base}$ and $Y_{novel}$ are disjoint. $D_{base}$ typically contains a large number of samples per base class while each novel class in $D_{novel}$ has only a few samples. $D_{base}$ can be used for pretraining a classification model before novel class samples are exposed to the model so that it can learn to distinguish between novel classes $Y_{novel}$ with only a few training samples.

In this paper, we consider the FSL setting where there exists prior knowledge about predefined visual similarities between classes. Specifically, any two classes $y_i$ and $y_j$ that are considered to be visually similar to each other are considered neighbors of each other. The set of such predefined pairs is denoted as the neighborhood set $P = \{(y_i, y_j)\}^n$. Note that a class pair outside of $P$ does not necessarily indicate that its two classes are dissimilar.

In the hematopoiesis cell classification problem, any two consecutive stage classes are considered to neighbors.

### 3.2. Pairwise Evidence Lower Bound

In this section we represent data using a graphical model that incorporates the similarity relationships between classes. Based on that, we derive the Pairwise Evidence Lower Bound (L-ELBO), a lower bound of the data pairwise likelihood on the graphical model.

We assume that each sample $x_{i_k}$ of class $y_i$ is generated from three factors: the generation parameter $\theta$, the class discriminative feature $c_{i_k}$ and the style $z_{i_k}$. The generation parameter $\theta$ is fixed and shared across all the observations of all classes. The class discriminative feature $c$ is constrained to be consistent among instances of the same class while instance-specific features are expressed via the style $z$. A sample $x_{i_k}$ is drawn from the distribution $p_\theta(x_{i_k}|c_{i_k}, z_{i_k})$. Let $X_i = \{x_{i_k}\}$ be the set of observations belonging to class $y_i$, and $Z_i$ be the corresponding set of styles. Since $X_i$ belongs to a single class $y_i$, all the observations of

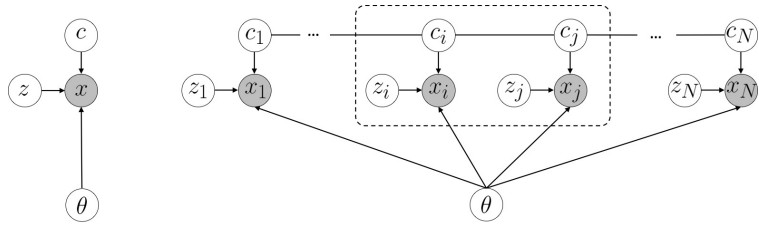

(a) MultilevelVAE          (b) RelationVAE

Figure 2: (a) The graphical model of MultilevelVAE and (b) An example of graphical model of RelationVAE.

the same class should share a single class discriminative feature $c_i$. This synthesis process is described in MultilevelVAE (Bouchacourt et al., 2017), with the graphical model illustrated at Figure 2a. MultilevelVAE is considered the baseline of our proposed method without incorporating class relationships.

We propose a graphical model which enforces connections between class discriminative feature variables $c_i$ to indicate the similarity relationships between classes as illustrated at Figure 2b. In this graphical model, class discriminative features of neighbor classes are connected to each other, meaning the likelihood of one class is dependent on its neighbor class. We solve for the generation parameters $\theta$ by maximizing the likelihood of the data $p_\theta(X) = p_\theta(X_1, X_2, X_3, ...)$ with respect to $\theta$. It is, however, complicated to derive the data likelihood due to the large number of possible neighborhood sets $P$. We instead maximize the pairwise likelihood $\mathbb{E}_{(y_i,y_j)\in P} \, p_\theta(X_i, X_j)$.

To this end, we first approximate the true posteriors $p_\theta(c_i|X_i, c_j)$, $p_\theta(c_j|X_i, X_j)$, $p_\theta(Z_i|X_i)$ by the Gaussian variational posteriors $q_{\phi_1}(c_i|X_i, c_j)$, $q_{\phi_2}(c_j|X_i, X_j)$, and $q_\gamma(Z_i|X_i)$. We also assume that the styles and the class discriminative features follow Gaussian distributions with mean 0 and unit covariance matrix.

To model the dependence of class $y_i$ on its neighbor class $y_j$, we define a transition probability from $y_j$ to $y_i$ as a Gaussian distribution with mean $c_j$ and learnable variance $\omega$: $P_\omega(c_i|c_j) = \mathcal{N}(c_i|\mu = c_j, \Sigma = \omega)$. We derive the log pairwise likelihood of the data from two neighbor classes $y_i$ and $y_j$ as follows: [1]

$$\log p_\theta(X_i, X_j) \geq \tag{1a}$$

$$\mathbb{E}_{q_\gamma(Z_i|X_i)} \mathbb{E}_{q_{\phi_2}(c_j|X_i,X_j)} \mathbb{E}_{q_{\phi_1}(c_i|X_i,c_j)} \log p_\theta(X_i|c_i, Z_i) \tag{1b}$$

$$+ \mathbb{E}_{q_\gamma(Z_j|X_j)} \mathbb{E}_{q_{\phi_2}(c_j|X_i,X_j)} \log p_\theta(X_j|c_j, Z_j) \tag{1c}$$

$$- \mathbb{E}_{q_{\phi_2}(c_j|X_i,X_j)} D_{KL}[q_{\phi_1}(c_i|X_i,c_j)||P_\omega(c_i|c_j)] \tag{1d}$$

$$- D_{KL}[q_{\phi_2}(c_j|X_i,X_j)||P(c_j)] \tag{1e}$$

$$- D_{KL}[q_\gamma(Z_i|X_i)||p(Z_i)] \tag{1f}$$

$$- D_{KL}[q_\gamma(Z_j|X_j)||p(Z_j)] \tag{1g}$$

---

1. Please see supplementary material for the proof

The RHS of eq. (1) is the pairwise evidence lower bound (P-ELBO) of $\log P(X_i, X_j)$ which we are going to maximize over the parameters $\theta$, $\phi_1$, $\phi_2$, $\gamma$, and $\omega$.

### 3.3. RelationVAE

In order to represent P-ELBO using a neural network, we first need to factorize $p_\theta(X_i|c_i, Z_i)$, $q_{\phi_1}(c_i|X_i, c_j)$, $q_{\phi_2}(c_j|X_i, X_j)$, and $q_\gamma(Z_i|X_i)$ into instance level quantities.

Similar to (Bouchacourt et al., 2017), we approximate any conditional Gaussian distribution $p(c|X)$, where $c$ is a single variable and $X$ is a set of observations, by a corresponding Group Evidence Accumulation function $GEA_{x_k \in X}[p(c_k|x_k)]$. $GEA$ is a Gaussian distribution with mean $\mu$ and covariance matrix $\Sigma$ defined as follows

$$\Sigma^{-1} = \sum_k \Sigma_k^{-1} \ , \ \mu^T\Sigma^{-1} = \sum_k \mu_k^T\Sigma_k^{-1} \tag{2}$$

where $\mu_k$ and $\Sigma_k$ are the mean and covariance matrix of each individual $p(c_k|x_k)$. Accordingly, $q_{\phi_1}(c_i|X_i, c_j)$ and $q_{\phi_2}(c_j|X_i, X_j)$ are approximated by $GEA_{x_{i_k} \in X_i}[q_{\phi_1}(c_{i_k}|x_{i_k}, c_j)]$ and $GEA_{x_{i_k} \in X_i, x_{j_l} \in X_j}[q_{\phi_2}(c_{j_k}|x_{i_k}, x_{j_l})]$ respectively.

Since samples in the same class are generated independently from each other, $p_\theta(X_i|c_i, Z_i)$ and $q_\gamma(Z_i|X_i)$ can be factorized into $\prod_{x_{ik} \in X_i} p_\theta(x_{ik}|c_i, z_{ik})$ and $\prod_{x_{ik} \in X_i} q_\gamma(z_{ik}|x_{ik})$.

To optimize the P-ELBO of the data, we propose RelationVAE (Fig. 3) with 3 encoders $\gamma$, $\phi_1$, $\phi_2$ and 1 decoder $\theta$, representing the functions with the same names discussed above. Specifically, $\gamma$ is used to encode a single sample into the intra-class variant. $\phi_1$ predicts the class information of a class from a sample of that class and the neighbor class information. The additional information from the neighbor class makes the prediction more robustly. $\phi_2$ predicts the class information of a class from its sample together with a sample of the neighbor class. $\theta$ is the decoder which generates samples of a class given its extracted class information and intra-class variants.

Since $\gamma$, $\phi_1$ and $\phi_2$ represent Gaussian distributions, their outputs are the means and covariance matrices diagonals of the distributions. From P-ELBO we get the loss functions:[2]

- From (1b) and (1c)

$$L_{recons} = \| x_i - x_i' \|^2 + \| x_j - x_j' \|^2 \tag{3}$$

- From (1d)

$$L_{content_1} = \frac{1}{2} \sum_k \left[ \exp(\log \Sigma_{c_i}^{(k)} - \log \omega^{(k)}) + \frac{(c_j^{(k)} - \mu_{c_i}^{(k)})^2}{\omega^{(k)}} + (\log \omega^{(k)} - \log \Sigma_{c_i}^{(k)}) \right] \tag{4}$$

- From (1e)

$$L_{content_2} = \frac{1}{2} \sum_k [\Sigma_{c_j}^{(k)2} + \mu_{c_j}^{(k)2} - 2\log(\Sigma_{c_j}^{(k)}) - 1] \tag{5}$$

---

2. Please see the supplementary material for the derivations

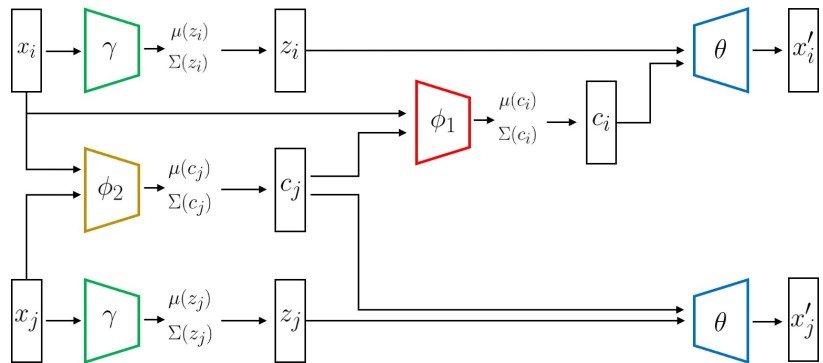

Figure 3: **RelationVAE Pipeline** (best viewed in color - same color networks share weights). The 3 encoders $\gamma$, $\phi_1$, $\phi_2$ encode observation $x_j$ from class $y_j$ and observation $x_i$ from its neighbor class $y_i$ into intra-class variant $z_i$, $z_j$ and the class information $c_i$, $c_j$. The decoder $\theta$ generates samples from the provided class information and intra-class variant.

- From (1f) and (1g)

$$L_{style} = \frac{1}{2} \sum_{e \in \{i,j\}} \sum_k [\Sigma_{z_e}^{(k)2} + \mu_{z_e}^{(k)2} - 2\log(\Sigma_{z_e}^{(k)}) - 1] \tag{6}$$

where $k$ is the shared index of dimensions on either the means $\mu$ and the variance $\Sigma$.

The P-ELBO loss function to train RelationVAE is the sum of the component losses:

$$L_{pELBO} = L_{recons} + L_{content_1} + L_{content_2} + L_{style} \tag{7}$$

To train RelationVAE, for each training iteration, we first sample a class $y_i$ and its visually similar neighbor $y_j$. Two corresponding batches $X_i$ and $X_j$, are sampled accordingly to train the network. When $y_i$ has no neighbor, we sample both batches $X_{i_1}$ and $X_{i_2}$ from $y_i$. The specific training algorithm is presented in Algorithm 1.

---

**for** *epoch* **do**
    sample a class $y_i$ and its neighbor $y_j$
    sample two batches $X_i$ and $X_j$ of size $K$
    **for** *k = 1..K* **do**
        calculate $q_{\phi_2}(c_{j_k}|x_{i_k}x_{j_k})$ and $q_\gamma(z_{j_k}|x_{j_k})$
        sample $z_{j_k} \sim q_\gamma(z_{j_k}|x_{j_k})$
    calculate $q_{\phi_2}(c_j|X_iX_j)$ from $\{q_{\phi_2}(c_{j_k}|x_{i_k}x_{j_k})\}_{k=1}^K$ using $GEA$
    sample $c_j \sim q_{\phi_2}(c_j|X_iX_j)$
    **for** *k = 1..K* **do**
        calculate $q_{\phi_1}(c_{i_k}|x_{i_k}c_j)$ and $q_\gamma(z_{i_k}|x_{i_k})$
        sample $z_{i_k} \sim q_\gamma(z_{i_k}|x_{i_k})$
    calculate $q_{\phi_1}(c_i|X_ic_j)$ from $\{q_{\phi_1}(c_{i_k}|x_{i_k}c_j)\}_{k=1}^K$ using $GEA$
    sample $c_i \sim q_{\phi_1}(c_i|X_ic_j)$
    **for** *k = 1..K* **do**
        calculate $p_\theta(x'_{i_k}|z_{i_k}c_i)$ and $p_\theta(x'_{j_k}|z_{j_k}c_j)$

    Update $\phi_1$, $\phi_2$, $\gamma$, $\omega$, and $\theta$ w.r.t. the loss function $L_{pELBO}$.
  **Algorithm 1:** Training RelationVAE

---

During testing, for a novel class $y_i$ with a neighbor class $y_j$, we predict its class discriminative feature $c_i$ by using $q_{\phi_1}(c_i|X_i, c_j)$. $c_i$ is then paired with randomly sampled $z_i$ to pass to $\theta$ from generate more data. Once the augmented data is generated, any classification method can be applied on these data.

## 4. Experiments

### 4.1. Hemapotoiesis dataset

We collected a dataset of 7433 cell images from 21 categories, representing blood cell development in various stages of maturation within the bone marrow. Each class corresponds to a cell type and its development phase. The data was collected from 19 whole slide images that were scanned under oil immersion at 100X magnification. A hematopathologist selected $2000 \times 2000$ pixel regions that contain mixtures of different cell types in various stages of development, where each cell was annotated as belonging to one of 21 commonly utilized hematopathology categories. We extracted $300 \times 300$ pixel patches around the labeled cells.

We split the dataset into two subsets: 12 base set classes and another 9 novel classes. The data is summarized in table 2.

### 4.2. Experimental Results

We conduct 1-shot and 5-shot experiments for 9-way classification (9 novel classes). For each experiment, we run 600 trials. In each trial we randomly sample a set of n novel classes together with the corresponding k training samples per class for n-way k-shot classification. We report top-1 accuracy scores (600 trial average) together with 95% confidence intervals.

We train RelationVAE on 1D visual features instead of images. The final classifier is also trained on the visual features generated by RelationVAE. Specifically, we use Baseline++ (Chen et al., 2019) with ResNet-18 backbone to extract features of input images to use with RelationVAE. For fair comparison, we also use the ResNet-18 backbone for all other compared methods.

We compare RelationVAE with the baseline MultilevelVAE (Bouchacourt et al., 2017) which does not incorporate relationships between classes. We also compare with SoTA FSL methods: MAML (Finn et al., 2017), RelationNet (Sung et al., 2018), Baseline++ (Chen et al., 2019), RFS (Tian et al., 2020), DSFN (Zhang and Huang, 2022), Distribution Calibration (Yang et al., 2021), DeltaGAN (Hong et al., 2022a).

The results on 1-shot and 5-shot learning are shown in table 1. As can be seen, the performance of our proposed RelationVAE is always better than that of Multilevel-VAE,. This proves our assumption that predicting class information from both observations and the neighbor class information is more robust than from observations only. Comparing to other FSL baselines, RelationVAE consistently achieves SoTA accuracy on 1-shot setting while its performance on 5-shot setting is better than most other methods. This again confirms that it is beneficial to use similarity relationships for knowledge transfer when training data is scarce.

Table 1: Few Shot Learning results with top-1 accuracy and the 95% confidence interval on Hematopoiesis dataset. The first and second blocks are discriminative methods and generative methods respectively.

| Method | 1-shot | 5-shot |
|---|---|---|
| MatchingNet (Vinyals et al., 2016) | 49.65 ± 0.4 | 68.90 ± 0.3 |
| MAML (Finn et al., 2017) | 55.12 ± 0.42 | 70.26 ± 0.51 |
| RelationNet (Sung et al., 2018) | 51.79 ± 0.2 | 68.89 ± 0.31 |
| Baseline++ (Chen et al., 2019) | 56.24 ± 0.51 | 68.27 ± 0.31 |
| RFS (Tian et al., 2020) | 55.89 ± 0.61 | 76.91 ± 0.21 |
| DSFN (Zhang and Huang, 2022) | 61.33 ± 0.47 | 81.91 ± 0.27 |
| MultilevelVAE (Bouchacourt et al., 2017) | 60.00 ± 0.5 | 74.70 ± 0.78 |
| Distribution Calibration (Yang et al., 2021) | 61.09 ± - | 81.32 ± - |
| Delta-GAN (Hong et al., 2022a) | 60.75 ± 0.36 | 80.59 ± 0.17 |
| RelationVAE | 62.84 ± 0.48 | 76.43 ± 0.35 |

## 5. Conclusion

In this paper, we study a few shot learning problem on Hematopoiesis data. We propose RelationVAE, a generative model incorporating neighborhood relationships between classes, which are available in Hematopoiesis. Our proposed RelationVAE is able to predict the class information not only from the few available training samples of that class but also from the class information of a neighbor class.Our experiments show the improved results of RelationVAE especially on the 1-shot setting where training data is extremely scarce.

## Acknowledgments

This research was partially supported by NSF grants IIS-2123920, IIS-2212046 and NCI grants NCI UH3CA225021, NCI U24CA215109.

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

## Appendix A.  Proof of Pairwise Evidence Lower Bound (P-ELBO)

In this section, we present the proof of Pairwise Evidence Lower Bound (P-ELBO) of the log likelihood of the data $X_i$ and $X_j$ which belong to two neighbor classes $y_i$ and $y_j$ (proof of equation (1) in the main paper). We first recapitulate the notations and assumptions we have for P-ELBO as below.

We assume the prior of class content and style variables be Gaussian distributions with zero mean and unit variance:

$$p(z) = \mathcal{N}(\mathbf{0}, \mathbf{I}), \ p(c) = \mathcal{N}(\mathbf{0}, \mathbf{I}) \tag{8}$$

Furthermore we approximate the true posteriors $p_\theta(c_i|X_i c_j)$, $p_\theta(c_j|X_i X_j)$, $p_\theta(Z|X)$ by the Gaussian variational posteriors $q_{\phi_1}(c_i|X_i c_j)$, $q_{\phi_2}(c_j|X_i X_j)$, and $q_\gamma(Z|X)$ whose distri-

butions are defined as follows:

$$q_{\phi_1}(c_i|X_i, c_j) = \mathcal{N}(c_i|\phi_1^\mu(X_i c_j), \phi_1^\Sigma(X_i c_j)) \tag{9}$$

$$q_{\phi_2}(c_j|X_i, X_j) = \mathcal{N}(c_j|\phi_2^\mu(X_i X_j), \phi_2^\Sigma(X_i X_j)) \tag{10}$$

$$q_\gamma(Z|X) = \mathcal{N}(Z|\gamma^\mu(X), \gamma^\Sigma(X)) \tag{11}$$

In order to enforce the dependence of a class $y_i$ on its neighbor $y_j$, we define a transition probability from $c_j$ to $c_i$ as a Gaussian distribution with mean $c_j$ and learnable variance $\omega$

$$P_\omega(c_i|c_j) = \mathcal{N}(c_i|\mu = c_j, \Sigma = \omega) \tag{12}$$

The log pairwise likelihood of the data $X_i$ and $X_j$ is derived as:

$$\log p_\theta(X_i, X_j) = \log \frac{p_\theta(X_i, X_j, c_i, c_j, Z_i, Z_j)}{p_\theta(c_i, c_j, Z_i, Z_j|X_i, X_j)}$$

$$= \log \frac{p_\theta(X_i, X_j|c_i, c_j, Z_i, Z_j)p_\omega(c_i, c_j)p(Z_i)p(Z_j)}{p_\theta(Z_i|X_i)p_\theta(Z_j|X_j)p_\theta(c_i|X_i, X_j, c_j)p_\theta(c_j|X_i, X_j)}$$

$$= \log \frac{p_\theta(X_i|c_i, Z_i)p_\theta(X_j|c_j, Z_j)p_\omega(c_i, c_j)p(Z_i)p(Z_j)}{p_\theta(Z_i|X_i)p_\theta(Z_j|X_j)p_\theta(c_i|X_i, X_j, c_j)p_\theta(c_j|X_i, X_j)}$$

$$+ \log \frac{q_\gamma(Z_i|X_i)}{q_\gamma(Z_i|X_i)} + \log \frac{q_\gamma(Z_j|X_j)}{q_\gamma(Z_j|X_j)} + \log \frac{q_{\phi_2}(c_j|X_i, X_j)}{q_{\phi_2}(c_j|X_i, X_j)} + \log \frac{q_{\phi_1}(c_i|X_i, c_j)}{q_{\phi_1}(c_i|X_i, c_j)} \tag{13}$$

Applying the expectation terms to $\log p_\theta(X_i X_j)$, we have:

$$\log p_\theta(X_i, X_j) = \mathbb{E}_{q_\gamma(Z_i|X_i)} \mathbb{E}_{q_\gamma(Z_j|X_j)} \mathbb{E}_{q_{\phi_2}(c_j|X_i, X_j)} \mathbb{E}_{q_{\phi_1}(c_i|X_i, c_j)} \log p_\theta(X_i, X_j)$$

$$= \mathbb{E}_{q_\gamma(Z_i|X_i)} \mathbb{E}_{q_{\phi_2}(c_j|X_i, X_j)} \mathbb{E}_{q_{\phi_1}(c_i|X_i, c_j)} \log p_\theta(X_i|c_i, Z_i)$$

$$+ \mathbb{E}_{q_\gamma(Z_j|X_j)} \mathbb{E}_{q_{\phi_2}(c_j|X_i, X_j)} \log p_\theta(X_j|c_j, Z_j)$$

$$- \mathbb{E}_{q_{\phi_2}(c_j|X_i, X_j)} D_{KL}[q_{\phi_1}(c_i|X_i, c_j)||P_\omega(c_i|c_j)]$$

$$- D_{KL}[q_{\phi_2}(c_j|X_i, X_j)||P(c_j)]$$

$$- D_{KL}[q_\gamma(Z_i|X_i)||p(Z_i)] \tag{14}$$

$$- D_{KL}[q_\gamma(Z_j|X_j)||p(Z_j)]$$

$$+ \mathbb{E}_{q_{\phi_2}(c_j|X_i, X_j)} D_{KL}[q_{\phi_1}(c_i|X_i, c_j)||p_\theta(c_i|X_i, c_j)]$$

$$+ D_{KL}[q_{\phi_2}(c_j|X_i, X_j)||p_\theta(c_j|X_i, X_j))]$$

$$+ D_{KL}[q_\gamma(Z_i|X_i)||p_\theta(Z_i|X_i)]$$

$$+ D_{KL}[q_\gamma(Z_j|X_j)||p_\theta(Z_j|X_j)]$$

which is equivalent to

$$\log p_\theta(X_i, X_j) - \mathbb{E}_{q_{\phi_2}(c_j|X_i,X_j)} D_{KL}[q_{\phi_1}(c_i|X_i,c_j)||p_\theta(c_i|X_i,c_j)] - D_{KL}[q_{\phi_2}(c_j|X_i,X_j)||p_\theta(c_j|X_i,X_j))]$$
(15a)

$$- D_{KL}[q_\gamma(Z_i|X_i)||p_\theta(Z_i|X_i)] - D_{KL}[q_\gamma(Z_j|X_j)||p_\theta(Z_j|X_j)]$$
(15b)

$$= \mathbb{E}_{q_\gamma(Z_i|X_i)} \mathbb{E}_{q_{\phi_2}(c_j|X_i,X_j)} \mathbb{E}_{q_{\phi_1}(c_i|X_i,c_j)} \log p_\theta(X_i|c_i, Z_i)$$
(15c)

$$+ \mathbb{E}_{q_\gamma(Z_j|X_j)} \mathbb{E}_{q_{\phi_2}(c_j|X_i,X_j)} \log p_\theta(X_j|c_j, Z_j)$$
(15d)

$$- \mathbb{E}_{q_{\phi_2}(c_j|X_i,X_j)} D_{KL}[q_{\phi_1}(c_i|X_i,c_j)||P_\omega(c_i|c_j)]$$
(15e)

$$- D_{KL}[q_{\phi_2}(c_j|X_i,X_j)||P(c_j)]$$
(15f)

$$- D_{KL}[q_\gamma(Z_i|X_i)||p(Z_i)]$$
(15g)

$$- D_{KL}[q_\gamma(Z_j|X_j)||p(Z_j)]$$
(15h)

This is the derivation of the Pairwise Evidence Lower Bound (P-ELBO) (equation (1) in the main paper).

## Appendix B. Derivations of RelationVAE losses

In this section, we present how loss functions of RelationVAE (equations (3, 4, 5, 6) in the main paper) are derived from P-ELBO (equation (1) in the main paper or equation (15) in section A).

### B.1. Derivation of $L_{recons}$ from equations (15c), (15d)

Similar to a traditional VAE, maximization of $P_\theta(x|c, z)$ is implemented by minimization of the reconstruction loss

$$\| x - x' \|^2$$
(16)

where $x$ is the input sample and $x'$ is the output of the decoder. As a result, from (15c) and (15d), we have the reconstruction loss $L_{recons}$ in equation (3) of the main paper.

$$L_{recons} = \| x_i - x_i' \|^2 + \| x_j - x_j' \|^2$$
(17)

### B.2. Derivation of $L_{content_1}$ from equation (15e)

For simplicity, we denote $\mu_{c_i}$ and $\Sigma_{c_i}$ as the mean and the diagonal of the diagonal covariance matrix of the distribution $q_{\phi_1}(c_i|X_i, c_j)$.

From (15e), we have $L_{content_1}$ defined as the KL Divergence between two multivariate normal distributions as follows:

$$D_{KL}[q_{\phi_1}(c_i|X_i,c_j)||P_\omega(c_i|c_j)] = D_{KL}[\mathcal{N}(\mu_{c_i},\Sigma_{c_i})||\mathcal{N}(c_j,\omega)]$$
$$= \frac{1}{2}\left(\text{Tr}(\omega^{-1}\Sigma_{c_i}) + (c_j-\mu_{c_i})^T\omega^{-1}(c_j-\mu_{c_i}) - K + \log\frac{\det\omega}{\det\Sigma_{c_i}}\right) \tag{18}$$

Considering each component, we have:

$$\text{Tr}(\omega^{-1}\Sigma_{c_i}) = \text{Tr}\begin{pmatrix} \frac{\Sigma_{c_i}^{(1)}}{\omega^{(1)}} & & & \\ & \frac{\Sigma_{c_i}^{(2)}}{\omega^{(2)}} & & \\ & & \ddots & \\ & & & \frac{\Sigma_{c_i}^{(K)}}{\omega^{(K)}} \end{pmatrix} = \sum_k \frac{\Sigma_{c_i}^{(k)}}{\omega^{(k)}} = \exp(\log\Sigma_{c_i}^{(k)} - \log\omega^{(k)}) \tag{19}$$

and

$$(c_j-\mu_{c_i})^T\omega^{-1}(c_j-\mu_{c_i}) = (c_j-\mu_{c_i})^T\begin{pmatrix} \frac{1}{\omega^{(1)}} & & & \\ & \frac{1}{\omega^{(2)}} & & \\ & & \ddots & \\ & & & \frac{1}{\omega^{(K)}} \end{pmatrix}(c_j-\mu_{c_i})$$
$$= \sum_k \frac{(c_j^{(k)}-\mu_{c_i}^{(k)})^2}{\omega^{(k)}} \tag{20}$$

and

$$\log\frac{\det\omega}{\det\Sigma_{c_i}} = \log\frac{\prod_k \omega^{(k)}}{\prod_k \Sigma_{c_i}^{(k)}}$$
$$= \log\prod_k \frac{\omega^{(k)}}{\Sigma_{c_i}^{(k)}} \tag{21}$$
$$= \sum_k (\log\omega^{(k)} - \log\Sigma_{c_i}^{(k)})$$

where $k$ is the index of dimensionality.

Putting equations 19, 20, and 21 together we have

$$D_{KL}[q_{\phi_1}(c_i|X_i,c_j)||P_\omega(c_i|c_j)] = \frac{1}{2}\sum_k\left[\exp(\log\Sigma_{c_i}^{(k)} - \log\omega^{(k)}) + \frac{(c_j^{(k)}-\mu_{c_i}^{(k)})^2}{\omega^{(k)}} + (\log\omega^{(k)} - \log\Sigma_{c_i}^{(k)}) - 1\right] \tag{22}$$

Skipping the constant 1 during optimization, equation (22) becomes $L_{content_1}$ (equation (4) in the main paper.)

### B.3. Derivation of $L_{content_2}$ from equation (15f) and $L_{style}$ from equations (15g),(15h)

Similar to the above, for simplicity, we denote $(\mu_{c_j}, \Sigma_{c_j})$, $(\mu_{z_i}, \Sigma_{z_i})$, $(\mu_{z_j}, \Sigma_{z_j})$ as means and diagonals of diagonal covariance matrices of the distributions $q_{\phi_2}(c_j|X_i, X_j)$, $q_\gamma(z_i|x_i)$ and $q_\gamma(z_j|x_j)$

From (15f), we have the loss $L_{content_2}$

$$
\begin{aligned}
L_{content_2} &= D_{KL}[q_{\phi_2}(c_j|X_i, X_j)||P(c_j)] \\
&= D_{KL}[\mathcal{N}(\mu_{c_j}, \Sigma_{c_j})||\mathcal{N}(\mathbf{0}, \mathbf{I})] \\
&= \frac{1}{2}\sum_k [\Sigma_{c_j}^{(k)2} + \mu_{c_j}^{(k)2} - 2\log(\Sigma_{c_j}^{(k)}) - 1]
\end{aligned}
\tag{23}
$$

Similarly for (15g) and (15h)

$$
D_{KL}[q_\gamma(z_i|x_i)||p(z_i)] = \frac{1}{2}\sum_k [\Sigma_{z_i}^{(k)2} + \mu_{z_i}^{(k)2} - 2\log(\Sigma_{z_i}^{(k)}) - 1]
\tag{24}
$$

$$
D_{KL}[q_\gamma(z_j|x_j)||p(z_j)] = \frac{1}{2}\sum_k [\Sigma_{z_j}^{(k)2} + \mu_{z_j}^{(k)2} - 2\log(\Sigma_{z_j}^{(k)}) - 1]
\tag{25}
$$

combining equations 24 and 25, we have the style loss $L_{style}$:

$$
L_{style} = \frac{1}{2}\sum_{e\in\{i,j\}}\sum_k [\Sigma_{z_e}^{(k)2} + \mu_{z_e}^{(k)2} - 2\log(\Sigma_{z_e}^{(k)}) - 1]
\tag{26}
$$

## Appendix C. Hematopoiesis dataset

In this section, we show the summary of the Hematopoiesis dataset at table 2.

## Appendix D. Additional Implementation Details

For simplicity, we train RelationVAE on 1D visual features instead of images. The final classifier is also trained on the visual features generated by RelationVAE. Specifically, we use Baseline++ (Chen et al., 2019) with ResNet-18 backbone to extract features of input images to use with RelationVAE. For fair comparison, we also use the ResNet-18 backbone for all other compared methods.

In our implementation of the RelationVAE, the encoder $\phi_1$ is a fully connected neural network which takes a sample $x_i$ of class $y_i$ and the class information $c_j$ of its neighbor class $y_j$ to output the mean and variance of the class discriminative feature $c_i$. This network consists of one intermediate fully connected layer with 500 output neurons and two fully connected output layers with 384 output neurons each, one for the mean and one for the variance. The other two encoders, $\gamma$ and $\phi_2$, are combined into a single fully connected

Table 2: Statistics of the Hematopoietis dataset

| Class | Base(B)/ Novel(N) | #Images | Class | Base(B)/ Novel(N) | #Images |
|---|---|---|---|---|---|
| Monocyte | B | 104 | Myelocyte | N | 378 |
| Early-Erythroblast | B | 332 | Platelet | N | 175 |
| Metamyelocyte | B | 245 | Intermediate-Erythroblast | N | 1537 |
| Plasmacell | B | 20 | Neutrophil | N | 702 |
| Band | B | 809 | RBC | N | 633 |
| Late-Erythroblast | B | 824 | MastCell | N | 7 |
| Promyelocyte | B | 143 | Eosinophil | N | 188 |
| Macrophage | B | 3 | Basophil | N | 88 |
| Lymphocyte | B | 430 | Proerythroblast | N | 126 |
| Blast | B | 82 | | | |
| Megakaryocyte | B | 398 | | | |
| NucleatedRBC | B | 196 | | | |

network which takes pairs of samples from neighbor classes $x_i$ and $x_j$ as input and outputs means and variances of the two styles $z_i$, and $z_j$ together with the mean and variance of the class discriminative feature $c_j$. Similar to $\phi_1$, this network consists of one intermediate fully connected layer with 500 output neurons, and six fully connected output layers for the means and variances of the styles (size 256) and class discriminative feature (size 384). Our decoder $\theta$ and the classifier are implemented as fully connected two-layer networks, each with 500 hidden neurons.

## Appendix E. Experiments on CUB and SDOGs

To show the generality of RelationVAE, we also test the method on two commonly used FSL datasets: Caltech USCD Birds (CUB) with 11,788 images of 200 bird classes (Wah et al., 2011) and Stanford Dogs (SDOGs) (Khosla et al., 2011) with 20,580 images of 120 dog categories selected from the ImageNet dataset. We split the datasets into base, validation and novel sets with the ratio 100:50:50 for CUB and 70:20:30 for SDOGs (see (Chen et al., 2019; Li et al., 2019)). Because human-provided relationship information is not included in these datasets, we simulate this information by utilizing the taxonomy hierarchy provided by (Chen et al., 2018) for CUB and the ImageNet hierarchy for SDOGs. Specifically we establish similarity relationships between classes under the same bird/dog family. Note that we are not using the hierarchies directly, but just employ them to simulate the human-provided relationships.

Results on the CUB and SDOGs datasets are shown in table 3. RelationVAE achieves SoTA performance on 1-shot setting on both datasets. On 5-shot setting, RelationVAE outperforms other methods on SDOGs dataset while being among the best methods on CUB dataset.

These experiments on CUB and SDOGs show that our method is general which can be applied on other FSL problems with predefined relationships.

Table 3: Few Shot Learning results with top-1 accuracy and the 95% confidence interval on the CUB and SDOGs datasets. The first block is discriminative methods, the second block is generative methods, and last block is our RelationVAE.

| Method | CUB | | SDOGs | |
|---|---|---|---|---|
| | 1-shot | 5-shot | 1-shot | 5-shot |
| Baseline++ (Chen et al., 2019) | 68.21 ± 0.94 | 80.44 ± 0.73 | 58.12 ± 1.00 | 72.97 ± 0.69 |
| MatchingNet (Vinyals et al., 2016) | 75.58 ± 0.95 | 85.48 ± 0.63 | 31.2 ± 0.40 | 76.4 ± 0.13 |
| MAML (Finn et al., 2017) | 75.9 ± 0.35 | 84.1 ± 0.12 | 61.9 ± 0.21 | 75.3 ± 0.12 |
| RelationNet (Sung et al., 2018) | 71.28 ± 0.98 | 83.52 ± 0.62 | 60.4 ± 0.41 | 75.1 ± 0.40 |
| RFS (Tian et al., 2020) | 75.7 ± 0.6 | 83.69 ± 0.3 | 59.8 ± 0.70 | 76.2 ± 0.54 |
| MultilevelVAE (Bouchacourt et al., 2017) | 76.3 ± 0.85 | 86.9 ± 0.58 | 56.25 ± 0.84 | 63.96 ± 0.75 |
| Distribution Calibration (Yang et al., 2021) | 78.8 ± - | 89.1 ± - | 69.8 ± - | 83.7 ± - |
| RelationVAE | 79.2 ± 0.87 | 87.8 ± 0.53 | 73.08 ± 0.84 | 84.27 ± 0.52 |

