# OpenReview forum: "Few Shot Hematopoietic Cell Classification"
_MIDL.io/2023/Conference — MIDL 2023 Poster_

### Official Review · Reviewer_5Wfd · 2023-01-29

**Confidence:** 4
**Preliminary Rating:** 4
**Recommendation:** Oral

**Summary:**

This paper proposes a VAE model to generate training data for hematopoiesis cell classes with limited training data. The main contribution is to model class relationships by transferring knowledge from visually similar classes. Visual similar classes are defined here as classes which represent neighbouring stages in the cell maturation process. A graphical model is proposed modelling this neighbourhood relationship and a pairwise ELBO is derived. The method is compared to several methods from the literature. All are trained on 1D features obtained from images and evaluated on a dataset of 7433 cell images from 21 categories obtained from 10 whole slide images.

**Strengths:**

- The paper is well written.
- Interesting approach to model the relationship between classes into a generative model. When there is prior knowledge of similarity relationships between classes, the idea to use this in a generative model is good and novel.
- The introduction and related work section are informative and well written.
- Data will be made public after acceptance.


**Weaknesses:**

- Information about the data and implementation/training is missing in the main text and only reported in the Appendix (without referencing it). These details are important to understand the experiments and results and should be put in the main text.
- The models are only trained with 1D features instead of the images. I don’t really understand why that is necessary (the authors say that this is for “simplicity”) and how the results would look like for 2D image models. One advantage of using 2D models would be the possibility to inspect the generated images for plausibility.
- The motivation is that images from underrepresented classes can be synthesized to increase the number of training images. But Table 2 (Appendix) shows, that the novel classes (for which images are to be synthesized) are quite large. And some base classes only have few samples. This sounds counterintuitive and needs to be explained.


**Deanonymize Review:**

no

**Detailed Comments:**

- From fig 1, I see some neighbouring cell types which look pretty different (Megakaryocyte and Platelet). Would it be possible to somehow quantify the expected similarity between classes?
- Also, I would suggest highlighting the cell types in fig 1, which are referenced to in the text (Myelocyte and Metamyelocyte, Late Erythroblast and Nucleated RBC).
- Related work: why are the relationships between those cells not integratable in these methods described in Sec 2, 2nd paragraph?
- How are the images divided into training, validation, and test set? Are they randomly divided, or is the information about the original whole-slide image considered? Meaning that images from the same whole-slide image are not mixed into training and testing.
- How are the baseline methods implemented and trained? Is the code publicly available and did they use default hyperparameters or where these tuned on the dataset used in this paper?
- Table 1: Why is Distribution Calibration (Yang et al., 2021) reported without std?


**Paper Type:**

methodological development

**Questions To Address In The Rebuttal:**

- Please rearrange some of the content to include more information about the dataset and experiments in the main text.
- Please train the model on 2D images to show at least a couple of generated data samples.
- Please motivate the selection of base and novel classes.

---

### Official Review · Reviewer_dDVL · 2023-02-03

**Confidence:** 2
**Preliminary Rating:** 5
**Recommendation:** Oral

**Summary:**

The paper proposes a few-shot learning method that takes into account the structural relationship between the classes. The method is evaluated on a hematopoiesis with 21 classes.

I find this type of problem (hierarchical class structure, where some of the classes are very rare) to be very relevant for histopathology images and more widely applicable. As such, the proposed methodology is very relevant for the field.

**Strengths:**

- Very relevant methodology, tackling a problem that is very common in real-world histopathology dataset.
- Well elaborated methodology and good experimental setup.
- Good improvements over baseline (for the 1-shot learning scenario).

**Weaknesses:**

As I mentioned before, I find this methodology more broadly applicable to histopathology imaging and I would have liked to see it applied to more than one dataset. For example, similar class hierarchies emerge in a variety of tumour subtype classification tasks, where some of the classes are very uncommon.



**Deanonymize Review:**

no

**Paper Type:**

both

**Questions To Address In The Rebuttal:**

While it is unlikely that this is feasible in during the short rebuttal period, my only major suggestion is to apply the same methodology to other histopathology datasets.

It would be also great to see some qualitative results in the appendix.

---

### Official Review · Reviewer_nuBc · 2023-02-08

**Confidence:** 4
**Preliminary Rating:** 4
**Recommendation:** Poster

**Summary:**

This paper proposes a VAE-based few-shot learning approach for the problem of hematopoietic cell classification in digital pathology via considering the structural relationship between the neighboring stages in the cell maturation process. Intuitively, the proposed method aims at generating more training samples for rare classes.

**Strengths:**

- Structure and language of the paper are sound.
- Intuitive motivation. The proposed method is well relevant to the problem where there exists prior knowledge about predefined visual similarities between classes.

**Weaknesses:**

- Unclear evidence of division between base and novel classes. In Table 2, some base class has few samples (Macrophage) while some novel class has many samples (Intermediate-Erythroblast).
- More experiments need to be done for comprehensive evaluation. For example, it would be necessary to evaluate the quality of generated samples. In addition to VAE, to my knowledge, there are also a number of SOTA GANs-based few-shot adaptation methods that are also trained on base categories and applied to novel categories [1-4].

[1] Zheng Gu, Wenbin Li, Jing Huo, Lei Wang, Yang Gao: "Lofgan: Fusing local representations for few-shot image generation." ICCV (2021)

[2] Guanqi Ding, Xinzhe Han, Shuhui Wang, Shuzhe Wu, Xin Jin, Dandan Tu, Qingming Huang: "Attribute Group Editing for Reliable Few-shot Image Generation." CVPR (2022)

[3] Yan Hong, Li Niu, Jianfu Zhang, Liqing Zhang: "Few-shot Image Generation Using Discrete Content Representation." ACM MM (2022)

[4] Yan Hong, Li Niu, Jianfu Zhang, Liqing Zhang: "DeltaGAN: Towards Diverse Few-shot Image Generation with Sample-Specific Delta." ECCV (2022)

**Deanonymize Review:**

no

**Paper Type:**

methodological development

**Questions To Address In The Rebuttal:**

More experimental evidence to validate the proposed method. More discussion regarding different Data-Synthesis-based FSL approaches (VAE, GANs, and even Diffusion models) can be provided in the related work.

---

### Official Review · Reviewer_ykjk · 2023-02-10

**Confidence:** 4
**Preliminary Rating:** 3

**Summary:**

This paper proposes a few-shot learning paradigm, namely RelationVAE, for hematopoietic cell classification. RelationVAE incorporerates visual similarity relationship between two consecutive maturation stage categories of hematopoietic cell to generate more robust samples for few-shot learning. The experiment result comparing with the baseline methods demonstrates the effectiveness of the proposed RelationVAE.

**Strengths:**

1. The idea of modeling relationship between two consecutive hematopoietic cell maturation stage categories based on the visual similarity is interesting.
2. The paper provides a detailed proof of the log pairwise likelihood of the data from two neighbor classes.

**Weaknesses:**

1. The paper is not easy to follow. The organization and writing of the paper need to be improved.
2. The clinical purpose and significance of few-shot hematopoietic cell classification need more explanation. It would be better if references were included as support for the argument in the second paragraph of the introduction.
3. The many symbols in figure 3 and equation 1 is difficult to find and correspond, making the proposed framework hard to understand. I would suggest the authors provide a more intuitive explanation in the caption of figure 3.
4. The baseline methods chosen for comparison are works from 2017–2020, which are no longer state-of-the-art. It does not seem solid to conclude that RelationVAE achieves SOTA performance. According to Table 3 in the appendix, no method can achieve 80% accuracy under a 1-shot setup for the CUB dataset. However, many works in 2021 and 2022 are capable of achieving 90%+ performance for the same task. (Refer to: https://paperswithcode.com/sota/few-shot-image-classification-on-cub-200-5-1)

**Deanonymize Review:**

no

**Detailed Comments:**

Please refer to the weaknesses.

**Paper Type:**

methodological development

**Questions To Address In The Rebuttal:**

1. The authors should provide more descriptions and references for the clinical purpose and significance of few-shot hematopoietic cell classification.
2. A more intuitive explanation of the components of RelationVAE should be provided.
3. The caption of figure 3 should provide more messages.
4. The experiment should include more recent works.
5. Why is there no standard deviation for distribution calibration results?

---

### Meta-Review · Area_Chair_29Sd · 2023-02-24

**Recommendation:** Accept (Poster)
**Confidence:** 5

**Metareview:**

With the revisions made by the authors, the paper has been substantially improved and the majority of the initial comments and concerns have been addressed. In particular, the comparison to DeltaGAN and DSFN was important in demonstrating the advantages of the proposed approach over recent state-of-the-art methods. The reviewers are generally positive with the submission. Consolidating the reviews and rebuttals, the meta-reviewer agrees with the assessment of the reviewers and would like to recommend acceptance of the paper.